# The RNAi Pathway Is Important to Control Mayaro Virus Infection in *Aedes aegypti* but not for *Wolbachia*-Mediated Protection

**DOI:** 10.3390/v12080871

**Published:** 2020-08-10

**Authors:** Pedro H. F. Sucupira, Álvaro G. A. Ferreira, Thiago H. J. F. Leite, Silvana F. de Mendonça, Flávia V. Ferreira, Fernanda O. Rezende, João T. Marques, Luciano A. Moreira

**Affiliations:** 1Mosquitos Vetores: Endossimbiontes e Interação Patógeno-Vetor, Instituto René Rachou – Fiocruz, Belo Horizonte, MG 30190-002, Brazil; psucupira@aluno.fiocruz.br (P.H.F.S.); alvaro.ferreira@fiocruz.br (Á.G.A.F.); smendonca@aluno.fiocruz.br (S.F.d.M.); fernanda.rezende@fiocruz.br (F.O.R.); 2Departamento de Bioquímica e Imunologia, Instituto de Ciências Biológicas, Universidade Federal de Minas Gerais, 6627-Pampulha-Belo Horizonte-MG, CEP 30270-901, Brazil; thjfl21@gmail.com (T.H.J.F.L.); fvianaferreira@gmail.com (F.V.F.); jtmarques2009@gmail.com (J.T.M.); 3Université de Strasbourg, CNRS UPR9022, Inserm U1257, 67084 Strasbourg, France

**Keywords:** *Alphavirus*, *Aedes aegypti*, Mayaro virus, innate immunity, RNAi pathway, *Wolbachia*

## Abstract

Mayaro virus (MAYV), a sylvatic arbovirus belonging to the *Togaviridae* family and *Alphavirus* genus, is responsible for an increasing number of outbreaks in several countries of Central and South America. Despite *Haemagogus janthinomys* being identified as the main vector of MAYV, laboratory studies have already demonstrated the competence of *Aedes aegypti* to transmit MAYV. It has also been demonstrated that the *Wolbachia*
*w*Mel strain is able to impair the replication and transmission of MAYV in *Ae. aegypti*. In *Ae. aegypti*, the small interfering RNA (siRNA) pathway is an important antiviral mechanism; however, it remains unclear whether siRNA pathway acts against MAYV infection in *Ae. aegypti*. The main objective of this study was to determine the contribution of the siRNA pathway in the control of MAYV infection. Thus, we silenced the expression of AGO2, an essential component of the siRNA pathway, by injecting dsRNA-targeting AGO2 (dsAGO2). Our results showed that AGO2 is required to control MAYV replication upon oral infection in *Wolbachia*-free *Ae. aegypti*. On the other hand, we found that *Wolbachia*-induced resistance to MAYV in *Ae. aegypti* is independent of the siRNA pathway. Our study brought new information regarding the mechanism of viral protection, as well as on *Wolbachia* mediated interference.

## 1. Introduction

Mayaro virus (MAYV) is an arthropod-borne *Alphavirus* that can infect humans and cause significant outbreaks in several regions of South America [1,2]. MAYV was first isolated in 1954 from febrile rural workers in Mayaro County, a southeastern region of Trinidad Island in the Caribbean [3]. Since then there have been numerous reports of human infections in several countries in Central and South America, usually in places with tropical forests, such as French Guiana, Bolivia, Peru, Suriname, Costa Rica, Guatemala, Venezuela, Mexico, Ecuador, Guyana, Panama, and Brazil [4].

MAYV is an *Alphavirus* from the Togaviridae family, closely related to Chikungunya (CHIKV), and produces a highly debilitating clinical illness characterized by high-grade fever, maculopapular skin rash and marked arthralgia that, in some patients can persist for months [1,2]. These clinical symptoms and manifestations are similar to both dengue and chikungunya fever and can easily be misdiagnosed leading to an underestimation of MAYV infection rates [1,4].

Along with Yellow Fever virus (YFV), MAYV seems to rely on a permanent enzootic sylvatic transmission cycle utilizing non-human primates and canopy-dwelling mosquitoes from *Haemagogus* genus [5,6,7]. Despite human infections being strongly associated with forest and rural environments representing spillovers from sylvatic cycle, several factors facilitate the possibility of MAYV urbanization that could lead to the emergence of permanent endemic urban cycles [8]. These include I) the regular occurrence of MAYV cases in cities located in neotropical regions [9] and, II) the high prevalence of *Aedes aegypti* and *Aedes albopictus*, both species presenting robust vector competence [10,11,12,13]. This potential of MAYV to urbanize and produce extensive epidemics in tropical urban areas have encouraged the search for effective strategies to block MAYV transmission in *Ae. aegypti*. Recently, several strains of the endosymbiont *Wolbachia pipientis* (Alphaproteobacteria: *Ricketsiales*) have been used in integrated arboviral control programs around the world [14,15,16,17]. On account of the fact that *Ae. aegypti* mosquitoes infected with *Wolbachia* have reduced vector competence for several arbovirus, namely Dengue virus (DENV) and CHIKV [18,19,20,21,22,23], recent laboratory studies demonstrated that this *Wolbachia*-induced resistance to RNA virus is also occurring for MAYV [24]. Therefore, the use of *Wolbachia*-infected mosquitoes may represent a strategy to control MAYV transmission.

*Wolbachia*-induced resistance to arbovirus infection in mosquitoes can provide us with a powerful approach to control mosquito-transmitted diseases [16,17] yet the cellular and molecular mechanisms that underlies the ability of *Wolbachia* to affect the replication of arboviruses are not well understood. Recent studies have hypothesized that this viral protection may be due to priming of major innate immune pathways, because they observed numerous immune genes to be upregulated in *Wolbachia*-infected versus uninfected mosquitoes [25]. Other hypothesis posits a role for competition between *Wolbachia* and viruses for host cellular resources, such as cholesterol [26,27]. It has also been suggested that the small interfering RNA (siRNA) pathway can have an effect on *Wolbachia*-mediated blocking phenotype in mosquito cells [28]. Conversely, there are other studies demonstrating that siRNA is not essential for *Wolbachia*-mediated antiviral protection in both *Drosophila melanogaster* (in vivo) and in *Ae. albopicuts* cell line (in vitro) [29,30].

Previous studies have elegantly demonstrated that siRNA is an important antiviral mechanism that is triggered by dsRNA produced in the host cells infected with virus and leads to cleavage of the viral RNA [31,32]. Recent studies support that siRNA also plays a role in mosquito antiviral immunity. In *Ae. aegypti* mosquitoes, siRNA has been shown to limit DENV [33], CHIK [34] and Sindbis virus (SINV) [35]. While the siRNA pathway can contribute to control viral replication, it was also demonstrated that this mechanism fails to efficiently silencing DENV in the midgut of *Ae. aegypti* mosquitoes [36]. Additionally, control of O’nyong-nyong virus (ONNV) replication in the midgut of *Anopheles gambiae* mosquitoes also appears to be independent of siRNA pathway [37]. These studies raise the possibility that siRNA pathway is not very efficient in preventing arbovirus replication in mosquito vectors.

Here we investigate the role of the siRNA pathway in controlling MAYV oral infection in *Ae. aegypti*. We show that AGO2, an essential component of the siRNA pathway, is required to control MAYV replication upon oral infection. Furthermore, we asked whether the siRNA pathway is required for *Wolbachia*-mediated blocking of MAYV in mosquitoes. We found that *Wolbachia*-induced resistance to MAYV in *Ae. aegypti* is independent of the siRNA pathway.

## 2. Materials and Methods

### 2.1. Mosquito Lineages

All experiments were carried out using two lines of *Ae. aegypti*. The first, called BR-BH, indicates wild Brazilian mosquitoes not infected with *Wolbachia*. This wild lineage was derived from material collected from ovitraps in the neighborhood of Venda Nova, Belo Horizonte—MG, Brazil. The second strain called *w*Mel-BR-BH was generated by introducing the *w*Mel strain of *Wolbachia* into the genetic background of Brazilian mosquitoes through backcrossing. Every five generations, 200 Brazilian BR-BH F1-F2 males for each 600 *w*Mel-BR-BH females were introduced into the cages of the *w*Mel-BR-BH colony to prevent inbreeding effects and maintain a similar genetic background between the two strains. The presence of *Wolbachia* in *w*Mel-BR-BH lineage was confirmed every generation by PCR, where it was observed 100% of infection across all generations.

### 2.2. Mosquito Rearing and Infections

*Ae. aegypti* mosquitoes were maintained in the insectary of the René Rachou Institute, Fiocruz, MG, Brazil at 28 °C and 70–80% relative humidity, in a 12:12 h light:dark photoperiod, and with 10% sucrose solution ad libitum. For infections through membrane feeding, 5–7 day old adult females were starved for 24 h and fed with a mixture of blood and virus supernatant containing MAYV, using a glass artificial feeding system covered with pig intestine membrane. After blood feeding, fully engorged females were selected and harvested individually for RNA extraction or dissection at different time points. Mosquitoes were ground in TRIzol (Invitrogen—Life Technologies, Carlsbad, CA, USA) using glass beads and total RNA was extracted from individual mosquitoes according to the manufacturer’s protocol.

### 2.3. Virus Propagation and Titration

Viral isolate of MAYV (TRVL 4675 strain), kindly supplied by the Flavivirus Laboratory of the Oswaldo Cruz Instituto—IOC / Fiocruz, Rio de Janeiro, Brazil), was propagated in C6/36 *Ae. albopictus* cells maintained on L15 medium supplemented with 10% FBS (fetal bovine serum—Gibco—Life Technologies), penicillin (Gibco—Life Technologies), streptomycin (Gibco—Life Technologies) and ciprofloxacin (Isofarma, Precabura, Eusébio, Ceará, Brazil). Briefly, cells were seeded to 70% confluency and infected at a multiplicity of infection (MOI) of 0.01. Cultures were maintained for 5 days at 28 °C when supernatant was collected. Virus stocks were kept at −80 °C before use. MAYV was titrated in Vero cells in 12-well tissue culture plates. We allowed the virus to adsorb for 1 h at 37 °C then an overlay of 0.4% agarose in DMEM with 2% FBS was added. Plates were incubated at 37 °C and 5% CO_2_ for 3 days. Then formaldehyde was added, and cells were covered with a crystal violet stain (70% water, 30% methanol, and 0.25% crystal violet) to visualize plaques.

### 2.4. Gene Silencing

RNA transcription was performed using T7/SP6 Megascript Kits (Ambion—Thermo Fisher Scientific—Life Technologies), following the manufacturer’s instructions. Briefly, template DNA containing both promoter sequences was obtained by RT–qPCR for dsAGO2, and by PCR amplification from a pGL3-Basic plasmid (Promega, Madison, WI, USA) containing the firefly luciferase sequence for dsFLUC. Primer sequences are provided in Appendix A. Adult 4 to 5-day-old females were intrathoracically injected with 69 nL of a dsRNA solution (7.2 µg µL^−1^) diluted in annealing buffer (20 mM Tris-HCl pH 7.5, 100 mM NaCl) using a nano-injector Nanoject III (Drummond Scientific, Broomall, PA, USA). Mosquitoes were allowed to recover for 48 h before feeding.

### 2.5. RT-qPCR

Total RNA was extracted from individual insects and treated with DNAse (Promega) according to the manufacturer’s protocol and then reverse transcribed using M-MLV reverse transcriptase (Promega). cDNA was subjected to quantitative PCR (qPCR) using the kit Power SYBR Green Master Mix (Applied Biosystems—Life Technologies), following the manufacturer’s instructions. Specific primers are listed in Appendix A.

### 2.6. Indirect Immunofluorescence Assays and Confocal Microscopy

Mosquitoes were anaesthetized with CO_2_ and then placed on ice-cold phosphate-buffered saline (PBS) solution (13 mM NaCl, 0.7 mM Na_2_HPO_4_, 1 mM NaH_2_PO_4_ at pH 7.2) during the dissection process. The dissected tissues were fixed in 4% paraformaldehyde diluted in PBS for 15 min, and then washed in PBS twice for 15 min, and incubated with 0.1% Triton-X-100 and 5% FBS in PBS (PTX-FBS) for 30 min at room temperature. Samples were then incubated overnight with primary antibody at 4°C. dsRNA-specific antibody (mouse monoclonal K1) was used at 1:200 dilution to detect viral dsRNA in the cell cytoplasm. Samples were washed with PTX-FBS, and then incubated in PTX-FBS with secondary antibodies conjugated with Alexa Fluor 546 (Molecular Probes, Eugene, Oregon, USA) for 1 h. Samples were then washed with PTX-FBS, and incubated with Alexa Fluor 488 Phalloidin and Hoechst 33342 stain solution (all by Molecular Probes) for 15 min. Then, samples were washed in PTX-FBS, dissected and mounted in Vectashield Mounting Medium for microscopy observation. Confocal images were taken with Laser Confocal Nikon C2+ microscopes (Nikon Healthcare, Tokyo, Japan) and processed in Fiji, version 1.53c [38].

### 2.7. Statistics

All statistical analyses were done using R, version 3.6.2, 2019-12-12 [39]. To compare infection rates between two groups a two tailed Fisher’s exact test was used, fisher.test in R. To compare infection rates (prevalence of infection) between more than two groups a chi-square test of independence was used, chisq.test in R. To compare the viral load between two groups we used a Mann-Whitney-Wilcoxon test, Wilcoxon.test in R. Multiple comparisons of viral loads were performed using a Kruskal-Wallis test, kruskal.test in R.

### 2.8. Ethics Statement

The human blood used in all experiments was obtained from a blood bank (Fundação Hemominas), according to the terms of an agreement with the René Rachou Institute, Fiocruz/MG (OF.GPO/CCO agreement—Nr 224/16).

## 3. Results

### 3.1. Characterization of MAYV Oral Infection in Aedes aegypti

MAYV is a single-stranded positive-sense RNA virus from Togaviridae family and is transmitted to humans by the bite of sylvatic mosquitoes such as *Hemagogus janthinomys*. However, recent laboratory studies reported that *Ae. aegypti* and *Ae. albopictus* can be infected and potentially transmit MAYV [13]. Here, to characterize the infection rates of MAYV (genotype D) in field-derived populations of *Ae. aegypti* mosquitoes (BR-BH), we initially analysed different concentrations of MAYV (genotype D) in the blood meal (Figure 1A,B). Analyzing the orally fed females seven days post feeding (d.p.f.), we found that a high dose of 10^9^ p.f.u. mL^−1^ of virus infects 97% of mosquitoes (Figure 1B). On the other extreme, using a dose of 10^3^ p.f.u. mL^−1^ of virus, we observed that only 43% of mosquitoes became infected. To test whether the infection rates of MAYV is dose-dependent, we compare the infection rates of all doses tested (10^3^, 10^5^, 10^7^, 10^8^, 10^9^ p.f.u. mL^−1^). A chi-square test of independence showed that there is a significant association between dose and infection rates, X2 (4, *n* = 150) = 41.4456, *p* = 2.173 × 10^−8^. Additionally, we tested the effect of *Wolbachia* presence on the MAYV infection rates using a population of mosquitoes carrying the *Wolbachia* strain *w*Mel originated from *D. melanogaster*. We first backcrossed the *w*Mel laboratory strain into the field/derived population BR-BH, to generate a population (*w*Mel-BR-BH) sharing the same genetic background. We observed that using 10^3^ p.f.u. mL^−1^ of virus no *w*Mel-BR-BH mosquitoes were infected, whereas using 10^9^ p.f.u. mL^−1^ 41% of the mosquitoes have detectable MAYV at 7 d.p.f. (Figure 1B). Similar to the mosquitoes without *w*Mel, the infection rates are dose-dependent, Chi-squared (4, *n* = 148) = 26.269, *p* = 2.793 × 10^−5^. Furthermore, for each MAYV dose we compared the infection rates of BR-BH mosquitoes with the *Wolbachia*-carrying mosquitoes (*w*Mel-BR-BH). The results show that for all different doses tested, *Wolbachia*-carrying mosquitoes were more resistant to MAYV infection than *Wolbachia*-free counterparts (Figure 1B).

To explore the effect of MAYV dose presented in the blood meal, we analyzed the viral load of the mosquitoes that tested positive at 7 d.p.f. We measured the MAYV RNA levels of single mosquitoes using reverse transcription quantitative PCR (RT-qPCR). In the *Wolbachia*-free mosquitoes (BR-BH), we found a significant association between dose and viral load, Kruskal-Wallis H test X2 (4, *n* = 109) = 32.969, *p* = 1.212 × 10^−6^. Similar effects are seen when comparing the viral load of *Wolbachia*-carrying mosquitoes (*w*Mel-BR-BH) across the different doses, Kruskal-Wallis H test X2 (2, *n* = 24) = 8.4091, *p* = 0.01493. However, we observed that all infected *w*Mel-BR-BH mosquitoes showed a reduced viral load when compared to the BR-BH mosquitoes, indicating that *Wolbachia* is not only reducing the amount of mosquitoes that become infected but also limiting the MAYV replication in the mosquitoes that presented an established infection at 7 d.p.f.

To further characterize MAYV infection in *Ae. aegypti*, we investigated virus tropism upon oral infection. Here we used an antibody specific to dsRNA, the monoclonal antibody K1, in immunofluorescence assays as proxy for viral replication. Previous studies have demonstrated that the K1 antibody is an efficient tool for the detection of virus dsRNA including Arenavirus, Echovirus, Poliovirus, Enterovirus, Rotavirus and Adenovirus [40,41]. Although endogenous dsRNA (not from viral origin) can be present in the cells, it is normally restricted to the nucleus [42,43]. Indeed, we were able to detect dsRNA in the cell cytoplasm of infected mosquitoes where MAYV is expected to replicate. We analyzed midgut epithelium, visceral muscle of the midgut, Malpighian tubules and thorax skeletal muscle of MAYV fed females and compared with blood-only fed females. Analysis of the midgut at 4 d.p.f. revealed the presence of dsRNA in the cytoplasm of the epithelium but not in the visceral muscle (Figure 2A,B). Despite the presence of dsRNA in the nucleus of several cells from the Malphigian tubules (probably of endogenous origin), we were unable to detected dsRNA in the cytoplasm of these cells (Figure 2C,D). Similarly, no dsRNA was detected in the cytoplasm of cells from the thorax skeletal muscle (Figure 2E,F). To explore that the presence of dsRNA in the cytoplasm of the midgut epithelium was due to the MAYV infection, we tested the infection rates of mosquitoes from the same batch used in the immunofluorescence assay. Since we used a high concentration (10^9^ p.f.u. mL^−1^) of MAYV in the infectious blood meal, we obtained an infection rate of 100%. Conversely, we were not able to detect MAYV in the mosquitoes fed with blood only. Thus, supporting the hypothesis that the dsRNA detected in the cell cytoplasm is from viral origin.

Overall, these immunofluorescence results suggest that at 4 d.p.f. the midgut epithelium as the only tissue, from those tested, infected with MAYV and undergoing replication. However, we cannot exclude the hypothesis that other tissues not analyzed in this study, such as fat body, could also be infected at 4 d.p.f.

### 3.2. siRNA Controls MAYV Replication in Aedes aegypti

To determine the contribution of the siRNA pathway in the control/clearance of MAYV, we injected dsRNA-targeting AGO2 (dsAGO2) to silence its expression in mosquitoes from the field-derived population (BR-BH). We selected the AGO2 gene since it is an essential component of the siRNA-induced silencing complex [31]. Two days after the dsAGO2 injection, mosquitoes were fed with an infective blood meal containing a dose of 10^9^ p.f.u. mL^−1^ of MAYV (Figure 3A). First, we quantified the reduction in the expression of AGO2 in mosquitoes that were microinjected with dsAGO2 by comparing it to the expression observed in mosquitoes injected with control dsRNA targeting the firefly luciferase sequence (dsFLUC). The result show that dsAGO2 injection led to a reduction of 67% of the mRNA amount found in mosquitoes compared to those injected with dsFLUC, Wilcoxon, *p* = 4 × 10^−12^ (Figure 3B). Since the dsAGO2 treated mosquitoes showed a significant reduction in mRNA, we then analysed the effects on MAYV infection at 4 and 7 d.p.f. Of note, we were not able to address the dsAGO2 treatment on infection prevalence due the fact that all mosquitoes became infected with MAYV in both groups, dsAGO2 and dsFLUC (Figure 3C). However, when we quantified the MAYV RNA levels presented in each mosquito at 4 d.p.f., we found that mosquitoes treated with dsAGO2 showed a significant increase in MAYV levels (Wilcoxon, *p* = 2.6 × 10^−8^; Figure 3D). Additionally, we also quantified the MAYV RNA levels in individual mosquitoes at 7 d.p.f., and similarly to 4 d.p.f., we observed that mosquitoes injected with dsAGO2 exhibited higher amounts of virus RNA when compared to dsFLUC treated mosquitoes (Wilcoxon, *p* = 4.3 × 10^−7^; Figure 3D). Taking into account that MAYV viremia in humans can range from 10^5^ to 10^7^ p.f.u. mL^−1^ [10], we have repeated the experiment using 10^7^ p.f.u. mL^−1^ of MAY in the infective blood meal. We observed that a reduction of 53% on the AGO2 had no significant effects on MAYV prevalence of infection at both 4 and 7 d.p.f. (Appendix A). Nonetheless, we observed that mosquitoes treated with dsAGO2 showed a significant increase in MAYV levels at 7 d.p.f. (Wilcoxon, *p* = 0.02306; Appendix A). Overall, our results show that AGO2 contributes to inhibit MAYV replication, indicating that the siRNA pathway controls MAYV infection in *Ae. aegypti*.

### 3.3. Wolbachia-Mediated MAYV Blocking in Aedes aegypti Is Independent of siRNA Pathway

Since previous studies demonstrated that the presence of *Wolbachia* in *Ae. aegypti* can block the replication of MAYV, we investigated whether the siRNA pathway contributes to the blocking phenotype caused by *Wolbachia* in *Ae. aegypti*. We evaluated the impact of siRNA on MAYV infection by silencing AGO2 in mosquitoes BR-BH carrying the *Wolbachia* strain *w*Mel (*w*Mel-BR-BH). We first injected dsAGO2 or dsFLUC, and then, two days later, mosquitoes were fed with an infective blood meal containing a dose of 10^9^ p.f.u. mL^−1^ of MAYV (Figure 4A). We confirmed the impairment of the siRNA pathway by quantifying the AGO2 mRNA in mosquitoes injected with dsAGO2 and compared to those injected with dsFLUC. Injection of dsAGO2 led to a 50% reduction of the AGO2 expression levels when compared to the dsFLUC treated mosquitoes (Wilcoxon, *p* = 1.3 × 10^−11^; Figure 4B).

We next assessed the effect of the mosquito siRNA pathway on *Wolbachia*-mediated MAYV protection by comparing the infection prevalence between mosquitoes treated with dsAGO2 and mosquitoes control treated with dsFLUC after an infectious blood meal (a dose of 10^9^ p.f.u. mL^−1^ of MAYV was used). We observed that *Wolbachia* protects AGO2 silenced mosquitoes in similar extent as control mosquitoes (dsFLUC), since both groups presented low infection rates (Figure 4C). This lack of interaction was detected for both time points 4 d.p.f (X2 (1, *n* = 54) = X, *p* = 1) and 7 d.p.f. (X2 (1, *n* = 45) = X *p* = 0.4959). Furthermore, we explored the role of siRNA *Wolbachia*-mediated MAYV protection by comparing individually the viral loads at 4 and 7 d.p.f. orally infected mosquitoes. Although the number of mosquitoes that became infected was small, the results show that mosquitoes treated with dsAGO2 developed similar MAYV levels to the control mosquitoes (Figure 4D). These similarities on the MAYV titers between the two mosquito groups (dsAGO2 and dsFLUC) were observed in both 4 and 7 d.p.f. (Wilcoxon, *p* = 0.65, and Wilcoxon, *p* = 0.41).

Additionally, we tested the effect of the mosquito siRNA pathway on *Wolbachia*-mediated MAYV protection using a lower dose of MAY (10^7^ p.f.u. mL^−1^) in the infective blood meal. We observed that a reduction of 50% on the AGO2 expression had no significant effect on MAYV prevalence of infection at both 4 and 7 d.p.f. (Appendix A). Although the number of mosquitoes that developed infection was limited, we observed that mosquitoes treated with dsAGO2 showed similar MAYV levels at 4 and 7 d.p.f. (Appendix A). Altogether, these results indicate that the siRNA pathway is not essential for *Wolbachia*-induced protection to MAYV infection in *Ae. aegypti*.

## 4. Discussion

In mosquitoes, like in most insects, the siRNA pathway plays a fundamental role in defense against viruses [33,35,44,45,46]. Here, we show that this pathway is also an important mechanism against MAYV infection in *Ae. aegypti*, a worldwide urban vector of several human arboviral diseases. We first experimentally analyze the vector competence of *Ae. aegypti* to MAYV. Our results show that under laboratory conditions, field-derived *Ae. aegypti* exhibited a high level of vector competence and that infection rates are dose dependent. This is in agreement with previous published data suggesting that *Ae. aegypti* could play a significant role in the transmission of MAYV [10,11,12,13]. Moreover, we also provided evidence that *Wolbachia* induces resistance to MAYV infection in *Ae. aegypti* and that this effect is also dose dependent. A previous study also showed that *Wolbachia* also protects against MAYV infection, indicating the *Wolbachia* strain *w*Mel present in *Ae. aegypti* mosquitoes can be also used to reduce the prevalence and severity of MAYV [24].

In mosquito vectors, the siRNA pathway is an import antiviral mechanism against DENV, YFV, CHIKV, O’nyong-nyong virus (ONNV) and Sindbis (SINV) virus [33,34,35,47,48]. However, the siRNA pathway in *Ae. aegypti* appears complex since recent studies demonstrated that this mechanism fails to efficiently silencing DENV in the midgut of infected mosquitoes [36]. Besides *Aedes*, in *Anopheles gambiae* mosquitoes ONNV replication in the midgut also appears to be independent of siRNA pathway [37]. To evaluate the role of siRNA in other mosquito-borne alphavirus, we tested whether the siRNA pathway is important to control MAYV infections in *Ae. aegypti*. Using a Brazilian field-derived population of *Ae. aegypti* we demonstrated that silencing the expression of a core component of the siRNA pathway, AGO2, is sufficient to increase the viral titers of MAYV. However, we were unable to detect any effect of AGO2 silencing on the infections rates, probably due to a limited reduction on AGO2 expression or simply by the fact that we used a high concentration of MAYV in the blood meal. Therefore, it will be interesting to unravelling whether lower amounts of MAYV ingested during the blood meal could be sufficient to observe an effect in the infection rates.

The *w*Mel strain of *Wolbachia* has been successfully introduced into *Ae. aegypti* mosquitoes and subsequently shown in laboratory studies to reduce transmission of a range of arbovirus that cause human diseases [18,19,20,21,22,23,24,49]. Furthermore, recent field trials demonstrated that deployment of the *w*Mel strain into local *Ae. aegypti* populations can result in a reduction of local dengue transmission [16,17]. Despite the effectiveness of *Wolbachia*’s viral protection, the mechanism(s) that underlies the ability of *Wolbachia* to control the virus replication is not well understood. Studies showed that *Wolbachia* can induce the production of reactive oxygen species (ROS); the modulation of immune pathways leading to an increase in the basal immune activity of the host; and competition for cellular resources essential for viruses, like cholesterol [18,25,50]. Despite the fact that siRNA pathway is one of the major antiviral mechanism, it appears to be less important to the *Wolbachia*-mediated protection to virus [30,51]. Conversely, a study performed in mosquito cells demonstrated that the RNAi pathway plays a small part in *Wolbachia*-mediated blocking of DENV [28]. To characterize the role of siRNA in the *Wolbachia*-mediated MAYV blocking phenotype, we silenced AGO2 in *Ae. aegypti* mosquitoes carrying *w*Mel. Our results demonstrated that *Wolbachia* protects AGO2-silenced mosquitoes to MAYV infection to the same extent as control mosquitoes (AGO2 not silenced). However, a minor increase in the infection rates was observed. Furthermore, will be important to speculate whether the strength of the AGO2 silencing (around 50%) could limit the scope of its interaction with the *Wolbachia*-mediated phenotype. We cannot rule out that remain expression of AGO2 is not sufficient to intermediate the *Wolbachia*-induced viral protection.

In summary, our results demonstrate that the siRNA pathway is important to control MAYV infections in *Ae. aegypti* mosquitoes, but not for *Wolbachia*-mediated protection. Although functional studies using gene silencing can be a great contribution to the understanding of complex biological pathways and associated phenotypes, it will be interesting to determine the role of the siRNA pathway using mosquitoes with loss-of-function mutations in genes from this pathway.

## Figures and Tables

**Figure 1 viruses-12-00871-f001:**
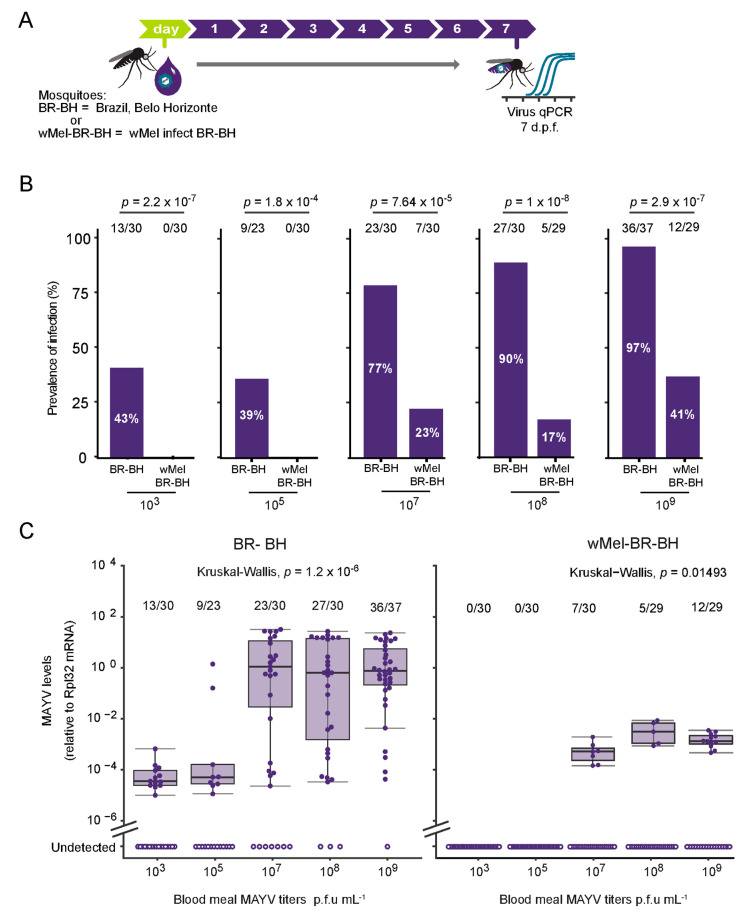
Characterization of MAYV oral infection in *Aedes aegypti* mosquitoes. (**A**) Scheme of the experimental design. BR-BH or *w*Mel-BR-BH 5–7 days old were fed on a blood meal containing 10^3^, 10^5^, 10^7^, 10^8^ or 10^9^ p.f.u. mL^−1^ of MAYV. Mosquitoes were collected at 7 d.p.f and tested individually by qPCR to detect viral RNA levels. (**B**) Prevalence of infection. Total number of mosquitoes tested are indicated above each column. Statistical analyses were performed using the two-tailed Fisher’s exact test. (**C**) MAYV RNA levels at 7 d.p.f. Each dot represents an individual whole mosquito. Bars in boxplot represent median viral load after log_10_ transformation. Upper and lower limit of the boxplot represent 75th and 25th percentile, respectively. Whiskers represent error bar using +/−1.5 × QR. Dots located outside the whiskers represent outliers. Statistical analyses were performed using the Kruskal-Wallis test, comparing the MAYV titers of the infected mosquitoes.

**Figure 2 viruses-12-00871-f002:**
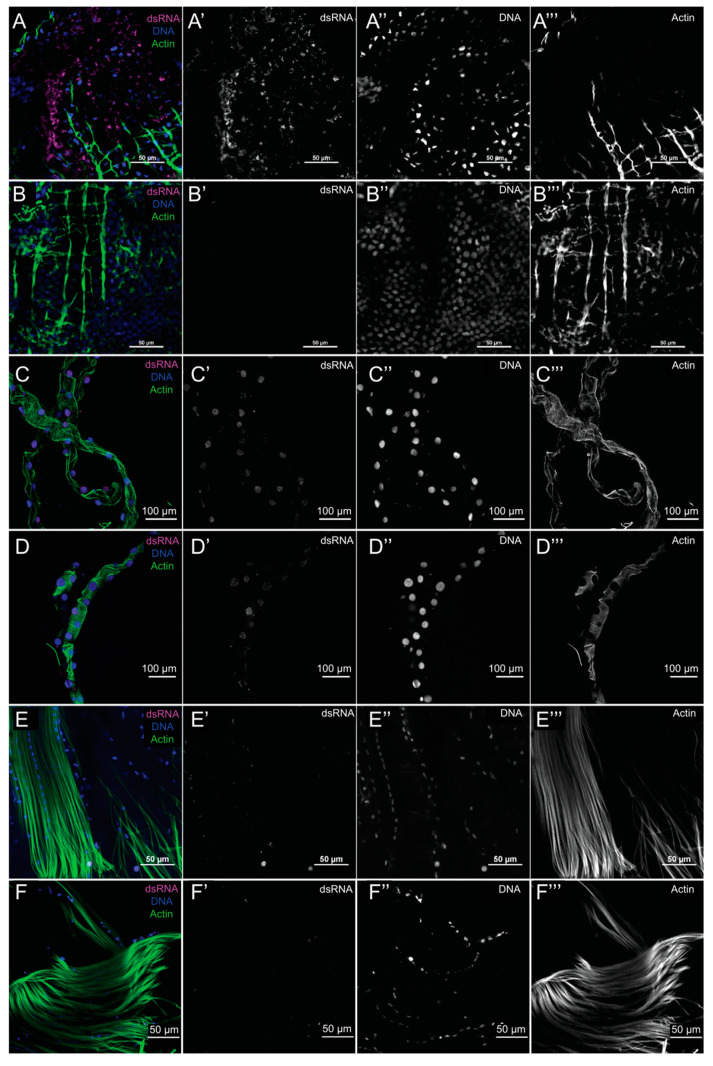
MAYV tissue tropism upon oral infection in *Aedes aegypti*. (**A**) MAYV is present in midgut epithelial enterocytes in BR-BH female mosquito 4 days after feeding on a blood meal containing 10^9^ p.f.u. mL^−1^ of MAYV. (**B**) Midgut from BR-BH female mosquito fed on blood only. (**C**) MAYV was not detected in the Malphigian tubules, in BR-BH female mosquito 4 days after feeding on a blood meal containing 10^9^ p.f.u. mL^−1^ of MAYV. (**D**) Malphigian tubules from BR-BH female mosquito fed on blood only. (**E**) MAYV was not detected in the thorax muscle in BR-BH female mosquito 4 days after feeding on a blood meal containing 10^9^ p.f.u. mL^−1^1 of MAYV. (**F**) Thorax muscle from BR-BH female mosquito fed on blood only. A, B, C, D, E, F, are images merging the triple-staining of the immunefluorescence assays of adult female tissues that were immunostained with antibody against dsRNA (magenta), actin marked with phalloidin (green) and DNA marked with Hoechst (blue). (**A’**), (**B’**), (**C’**), (**D’**), (**E’**), (**F’**) correspond to dsRNA immunostaining. (**A’’**), (**B’’**), (**C’’**), (**D’’**), (**E’’**), (**F’’**) correspond to DNA staining. (**A’’’**), (**B’’’**), (**C’’’**), (**D’’’**), (**E’’’**), (**F’’’**) correspond to actin staining.

**Figure 3 viruses-12-00871-f003:**
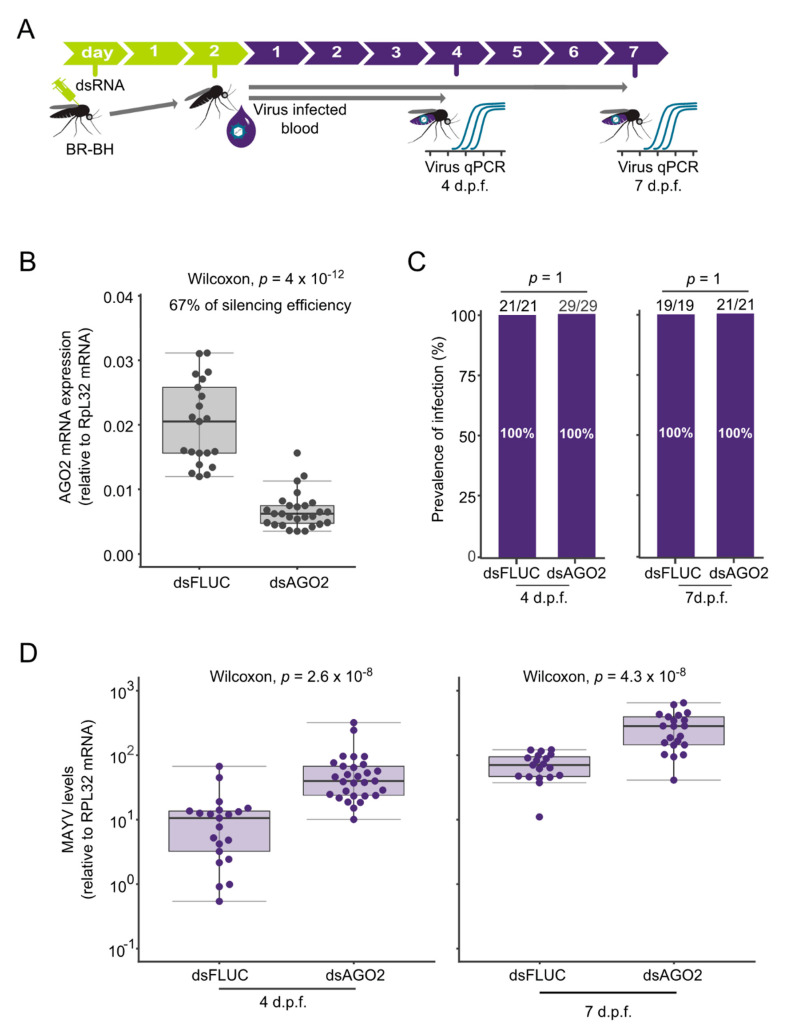
Knockdown of AGO2 increases MAYV replication in *Aedes aegypti.* (**A**) Scheme of the silencing process using dsRNA. BR-BH females were intrathoracically injected with dsAGO2 and dsFLUC (control group). 2 days post microinjection, mosquitoes were fed on a blood meal containing 10^9^ p.f.u. mL^−1^ of MAYV. Mosquitoes were collected at 4 and 7 d.p.f and tested individually by qPCR to detect viral RNA levels. (**B**) Mosquitoes collected at 4 d.p.f were also tested individually by qPCR for AGO2 mRNA expression for measuring the silencing efficiency. Each dot represents an individual whole mosquito. Statistical analyses were performed using the Mann-Whitney-Wilcoxon test, comparing the Ago2 expression levels. (**C**) Prevalence of infection. Total number of mosquitoes tested are indicated above each column. Statistical analyses were performed using the two-tailed Fisher’s exact test. (**D**) MAYV RNA levels at 4 and 7 d.p.f. Each dot represents an individual whole mosquito. Statistical analyses were performed using the Mann-Whitney-Wilcoxon test, comparing the MAYV titers of the infected mosquitoes.

**Figure 4 viruses-12-00871-f004:**
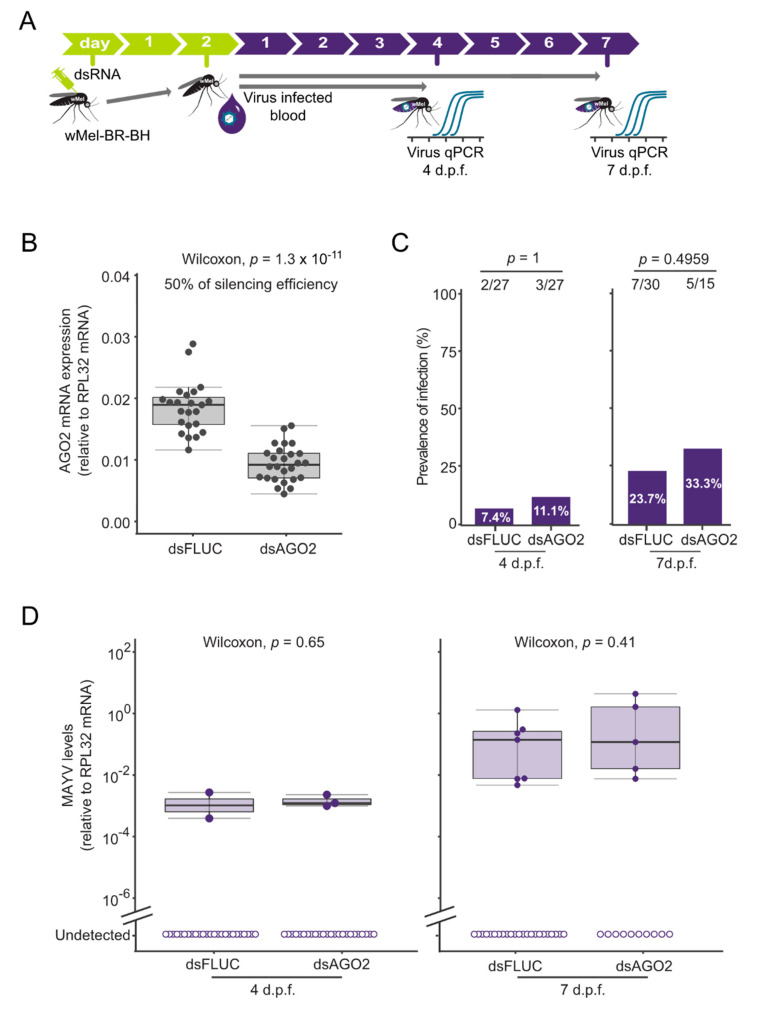
*Wolbachia*-mediated MAYV blocking in *Aedes aegypti* is independent of siRNA pathway. (**A**) Scheme of the silencing process using dsRNA. *w*Mel-BR-BH females were intrathoracically injected with dsAGO2 and dsFLUC (control group). 2 days post microinjection, mosquitoes were fed on a blood meal containing 10^9^ p.f.u. mL^−1^ of MAYV. Mosquitoes were collected at 4 and 7 d.p.f and tested individually by qPCR to detect viral RNA levels. (**B**) Mosquitoes collected at 4 d.p.f were also tested individually by qPCR for AGO2 mRNA expression for measuring the silencing efficiency. Each dot represents an individual whole mosquito. Statistical analyses were performed using the Mann-Whitney-Wilcoxon test, comparing the AGO2 expression levels. (**C**) Prevalence of infection. Total number of mosquitoes tested are indicated above each column. Statistical analyses were performed using the two-tailed Fisher’s exact test. (**D**) MAYV RNA levels at 4 and 7 d.p.f. Each dot represents an individual whole mosquito. Statistical analyses were performed using the Mann-Whitney-Wilcoxon test, comparing the MAYV titers of the infected mosquitoes.

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
