# Peer review of "The RNAi Pathway Is Important to Control Mayaro Virus Infection in Aedes aegypti but not for Wolbachia-Mediated Protection"

_viruses, 2020, doi:10.3390/v12080871_

Round 1

Reviewer 1 Report

This is an interesting and important study. Mosquitoes are important vectors of infectious diseases and the relation with Wolbachia seems still elusive. This study shows that Wolbachia reduces the viral genome number in infected mosquitoes. As with most arboviruses, Mayaro infection of mosquitoes triggers an RNA interference response. The study shows that this response is antiviral and it does not play an important role in the antiviral properties of Wolbachia. The studies are logically presented and the generally the manuscript is well-written. I enjoyed reading it. Methods It is noted that PCR is the method used for titrating virus in mosquitoes.

Is there any reason why the authors do not do a plaque assay or a TCID50? The authors might like to consider and perhaps comment on this.

Is there any reason why the authors picked the 109 pfu/ml as an infectious dose? It seems excessive considering a vaerimic sample usually has 1 x 107- 5 x 107 pfu/ml of blood. The authors might want to comment on this.

The number of mosquitoes per experiment seems sufficient, especially in injected mosquitoes since survival can be difficult sometimes. Results The paper characterizes the Mayaro infection of Aedes aegypti mosquitoes with or without Wolbachia. The authors measure virus genomes at day 7 after consuming blood meals containing different virus titres. Mosquitoes carrying Wolbachia show no presence of virus at early time points and reduced viral RNA levels at later time points. This is a relatively novel finding with Mayaro virus.

In figure 2 the authors use a dsRNA antibody to locate the presence of the virus in various tissues in infected mosquitoes. The authors try to show the distribution of the virus within the mosquito. Mayaro seems to infect midgut cells but not Malpighian tubes or thorax muscle. The nuclear k1 dsRNA staining is bizarre. Sometimes caught mosquito lines can carry some endogenous virus RNA viruses, hence the dsRNA in the nucleus. Otherwise I would like the authors to elaborate or give examples/reference of dsRNA staining in the nucleus of insect cells. In addition, if the authors want to include this figure they should show mosquitoes with Wolbachia and how that affects viral distribution, perhaps at a later time point (day 7).

RNAi is an important antiviral mechanism is mosquitoes and the authors show this by inhibiting a key molecule in the RNAi pathway through silencing. The partial silencing results are understandable since you are silencing the silencing machinery, therefore it is almost impossible to completely shut it down. The difference in RNA levels is however sufficient to show the relevance of this pathway in figure 3. The author’s show that inhibition of the RNAi pathway is not enough to overcome the presence of Wolbachia by the virus. The differences between AGO2 dsRNA and control dsRNA mosquitoes are not significant. The numbers and values are low so Wolbachia probably mask any increase of Mayaro in AGO2 dsRNA treated mosquitoes as the authors point out.

Discussion The discussion is brief but compares the findings to other studies. It seems this is one of the first papers characterizing the role of RNAi in Wolbachia infected mosquitoes. Other papers have used mostly cell lines. The finding of Mayaro being inhibited by Wolbachia quite efficiently is also relevant and novel. The conclusions are well founded on the results. Wolbachia and Aedes aegypiti/albopictus should be in Italics in the whole text

Author Response

Reviewer 1

Comments and Suggestions for Authors

This is an interesting and important study. Mosquitoes are important vectors of infectious diseases and the relation with Wolbachia seems still elusive. This study shows that Wolbachia reduces the viral genome number in infected mosquitoes. As with most arboviruses, Mayaro infection of mosquitoes triggers an RNA interference response. The study shows that this response is antiviral and it does not play an important role in the antiviral properties of Wolbachia. The studies are logically presented and the generally the manuscript is well-written. I enjoyed reading it. Methods It is noted that PCR is the method used for titrating virus in mosquitoes.

Is there any reason why the authors do not do a plaque assay or a TCID50? The authors might like to consider and perhaps comment on this.

R: Thank you for your positive comments. The PCR method is the chosen methodology in our laboratory used to quantify virus levels (through RT-qPCR) since we do not have a cell culture laboratory to frequently perform plaque assays. However, in the beginning of this study, we compared the plaque assay with PCR and observed that both methods were reliable to quantify MAYV. Please see the figure below and I let the Editor decide whether it would be appropriate to include this as a supplemental material.

Figure S3: Simple linear regression of viral quantification by plaque assay and RT-qPCR during MAYV infection. The graphs show the relationship between the viral load measured by RT-qPCR and p.f.u/mL per mosquito. Each blue dot represents an individual female mosquito. The line represents the trend between the points.

Is there any reason why the authors picked the 109 pfu/ml as an infectious dose? It seems excessive considering a vaerimic sample usually has 1 x 107- 5 x 107 pfu/ml of blood. The authors might want to comment on this.

R: Thank you for pointing this out. We do agree with the reviewer and we have included a new experiment replicate done with a lower infectious dose 107 pfu/ml. Please see lines 251–258, 295–302 and supplementary figures 1 and 2.

The number of mosquitoes per experiment seems sufficient, especially in injected mosquitoes since survival can be difficult sometimes. Results The paper characterizes the Mayaro infection of Aedes aegypti mosquitoes with or without Wolbachia. The authors measure virus genomes at day 7 after consuming blood meals containing different virus titres. Mosquitoes carrying Wolbachia show no presence of virus at early time points and reduced viral RNA levels at later time points. This is a relatively novel finding with Mayaro virus.

In figure 2 the authors use a dsRNA antibody to locate the presence of the virus in various tissues in infected mosquitoes. The authors try to show the distribution of the virus within the mosquito. Mayaro seems to infect midgut cells but not Malpighian tubes or thorax muscle. The nuclear k1 dsRNA staining is bizarre. Sometimes caught mosquito lines can carry some endogenous virus RNA viruses, hence the dsRNA in the nucleus. Otherwise I would like the authors to elaborate or give examples/reference of dsRNA staining in the nucleus of insect cells. In addition, if the authors want to include this figure they should show mosquitoes with Wolbachia and how that affects viral distribution, perhaps at a later time point (day 7).

R: Thank you for pointing this out. Indeed, some wild-caught lineages from our laboratory tested positive for insect specific virus, namely Phasi charoen-like virus (PCLV). However, there is a line of evidence that the dsRNA detected in the nucleus host cells is endogenous dsRNA expressed by the host genome rather from viral genomes (please see references below). We do recognize that the description to the K1 antibody was not very clear. To address this, we made some changes in the text; please see line 202 to 2012. Regarding the Wolbachia effect on MAY tissue distribution on Aedes aegypti, we do agree that is a very important topic to explore in detailed. However, this topic was not on the scope of this manuscript.

Mateer E, Paessler S, Huang C. Confocal Imaging of Double-Stranded RNA and Pattern Recognition Receptors in Negative-Sense RNA Virus Infection. J Vis Exp. 2019;(143):10.3791/59095. Published 2019 Jan 26. doi:10.3791/59095

Richardson SJ, Willcox A, Hilton DA, et al. Use of antisera directed against dsRNA to detect viral infections in formalin-fixed paraffin-embedded tissue. J Clin Virol. 2010;49(3):180-185. doi:10.1016/j.jcv.2010.07.015

RNAi is an important antiviral mechanism is mosquitoes and the authors show this by inhibiting a key molecule in the RNAi pathway through silencing. The partial silencing results are understandable since you are silencing the silencing machinery, therefore it is almost impossible to completely shut it down. The difference in RNA levels is however sufficient to show the relevance of this pathway in figure 3. The author’s show that inhibition of the RNAi pathway is not enough to overcome the presence of Wolbachia by the virus. The differences between AGO2 dsRNA and control dsRNA mosquitoes are not significant. The numbers and values are low so Wolbachia probably mask any increase of Mayaro in AGO2 dsRNA treated mosquitoes as the authors point out.

Discussion The discussion is brief but compares the findings to other studies. It seems this is one of the first papers characterizing the role of RNAi in Wolbachia infected mosquitoes. Other papers have used mostly cell lines. The finding of Mayaro being inhibited by Wolbachia quite efficiently is also relevant and novel. The conclusions are well founded on the results. Wolbachia and Aedes aegypiti/albopictus should be in Italics in the whole text.

R: Thank you for the comments. We have revised the whole paper and changed Aedes aegypti/albopictus to Italics.

Reviewer 2 Report

The paper is an interesting discussion of the role of the RNAi pathway in Wolbachia infected mosquitoes, and whether this is important for the reduction of mosquito infections in Wolbachia infected mosquitoes.

The paper is very nicely presented and was a pleasure to review so thank you for the effort put in to this effort. Although this is an intriguing topic and the paper is well written, I have a couple fo concerns, in particular, two major comments and several minor comments.

Major comment.

For figure 3. The use of 10^9 of MAYV while I understand the use of the high titer, this prevents you from observing how important RNAi is in MAYV infection. Without this, you don't have sufficient evidence to identify how important RNAi is for the Wolbachia. It would need to be used with lower titers to show increased infection in the RNAi, and again in the Wolbachia infected mosquitoes. This I think is an important experiment, as the lower titers will allow you to identify the  relative importance of the RNAi vs Wolbachia in reducing/increasing MAYV infection. I think using 10^5 and 10^ 3 would give you some information. 

Line 199- Did you confirm this using dissection and qPCR with other mosquitoes? Given the lack of specificity of the dsRNA that you are using, confirmation with a qPCR from dissected material would increase the robustness of this approach. In addition, use of a MAYV antibody against the E2 protein would make this much more robust. Although I recognise this is outside the scope of this investigation. 

Minor Comments

Line 87 - How long have the mosquitoes been colonised.

Also did you confirm that they all had WOlbachia? _ this should be included.

Line 157 - Italicise Haemogogus janthinomys

Line 164 - The use fo MAYV at 10^9 is not necessarily biological relevant. This should be addressed. 

Line 173 - please use Chi-squared rather than X2.

Author Response

Reviewer 2

Comments and Suggestions for Authors

The paper is an interesting discussion of the role of the RNAi pathway in Wolbachia infected mosquitoes, and whether this is important for the reduction of mosquito infections in Wolbachia infected mosquitoes.

The paper is very nicely presented and was a pleasure to review so thank you for the effort put in to this effort. Although this is an intriguing topic and the paper is well written, I have a couple fo concerns, in particular, two major comments and several minor comments.

Major comment.

For figure 3. The use of 10^9 of MAYV while I understand the use of the high titer, this prevents you from observing how important RNAi is in MAYV infection. Without this, you don't have sufficient evidence to identify how important RNAi is for the Wolbachia. It would need to be used with lower titers to show increased infection in the RNAi, and again in the Wolbachiainfected mosquitoes. This I think is an important experiment, as the lower titers will allow you to identify the  relative importance of the RNAi vs Wolbachia in reducing/increasing MAYV infection. I think using 10^5 and 10^ 3 would give you some information. 

R: Thank you for your comments. As mentioned above, we agree with the reviewer and have included a new experiment replicate done with a lower infectious dose 107 pfu/ml. Please see lines 251–258, 295–302 and supplementary figures 1 and 2.

Line 199- Did you confirm this using dissection and qPCR with other mosquitoes? Given the lack of specificity of the dsRNA that you are using, confirmation with a qPCR from dissected material would increase the robustness of this approach. In addition, use of a MAYV antibody against the E2 protein would make this much more robust. Although I recognise this is outside the scope of this investigation. 

R: Thank you for your comments. We do agree that is important to have the RTqPCR confirmation of the infection rates of the mosquitos used in the immunofluorescence assay. To address that we did the RT-qPCR to mosquitoes from the same same batch used in the immunofluorescence assay and observed an infection rate of 100%. We add to the manuscript text the corresponding information; please see lines 213 to 218.

Minor Comments

Line 87 - How long have the mosquitoes been colonised.

R: Thank you for pointing this out. In this paper we used mosquitoes from F4-F5 generations.

Also did you confirm that they all had Wolbachia? _ this should be included.

R: Thank you for pointing this out. We do agree that is very important to confirm that the mosquitoes from the wMel-BR-BH population are infected with Wolbachia across the generations. Indeed, in our laboratory we confirm by PCR the presence of Wolbachia in every single generation, this information was added to the manuscript text, please see line 94 and 95.

Line 157 - Italicise Haemogogus janthinomys

R: Thank you for pointing this out. We have changed to Italics.

Line 164 - The use fo MAYV at 10^9 is not necessarily biological relevant. This should be addressed. 

R: Thank you for pointing this out. We do agree with the reviewer and we have included a new experiment replicate done with a lower infectious dose 107 pfu/ml. Please see lines 251–258, 295–302 and supplementary figure 1 and 2.

Line 173 - please use Chi-squared rather than X2.

R: Thank you for pointing this out. We changed it as suggested.

Round 2

Reviewer 1 Report

The authors addressed the comments satisfactorily 

Reviewer 2 Report

The authors have revised the manuscript and repeated the experiments that I thought were required.

I have no major issues with the paper.